# Respectful maternity care delivered within health facilities in Bangladesh, Ghana and Tanzania: a cross-sectional assessment preceding a quality improvement intervention

Alexander Manu [1,2] Nabila Zaka,[3] Christina Bianchessi,[3] Edward Maswanya,[4] John Williams,[5] Shams E Arifeen[6]

For numbered affiliations see end of article.

**Correspondence to**
Dr Alexander Manu;
makmanu128@gmail.com

## ABSTRACT

**Objective** To assess respectful maternity care (RMC) in health facilities.

**Design** Cross-sectional study.

**Setting** Forty-three (43) facilities across 15 districts in Bangladesh, 16 in Ghana and 12 in Tanzania.

**Participants** Facility managers; 325 providers (nurses/midwives/doctors)—Bangladesh (158), Ghana (86) and Tanzania (81); and 849 recently delivered women—Bangladesh (295), Ghana (381) and Tanzania (173)—were interviewed. Observation of 641 client–provider interactions was conducted—Bangladesh (387), Ghana (134) and Tanzania (120).

**Assessment** Trained social scientists and clinicians assessed infrastructure, policies, provision and women's experiences of RMC (emotional support, respectful care and communication).

**Primary outcome** RMC provided and/or experienced by women.

**Results** Three (20%) facilities in Bangladesh, four (25%) in Ghana and three (25%) in Tanzania had no maternity clients' toilets and one-half had no handwashing facilities. Policies for RMC such as identification of client abuses were available: 81% (Ghana), 73% (Bangladesh) and 50% (Tanzania), but response was poor. Ninety-four (60%) Bangladeshi, 26 (30%) Ghanaian and 20 (25%) Tanzanian providers were not RMC trained. They provided emotional support during labour care to 107 (80%) women in Ghana, 95 (79%) in Tanzania and 188 (48.5%) in Bangladesh, and were often courteous with them—236 (61%) in Bangladesh, 119 (89%) in Ghana and 108 (90%) in Tanzania. Due to structural challenges, 169 (44%) women in Bangladesh, 49 (36%) in Ghana and 77 (64%) in Tanzania had no privacy during labour. Care was refused to 13 (11%) Tanzanian and 2 Bangladeshi women who could not pay illegal charges. Twenty-five (7%) women in Ghana, nine (6%) in Bangladesh and eight (5%) in Tanzania were verbally abused during care. Providers in all countries highly rated their care provision (95%–100%), and 287 (97%) of Bangladeshi women, 368 (97%) Ghanaians and 152 (88%) Tanzanians reported 'satisfaction' with the care they received. However, based on their facility experiences, significant (p<0.001) percentages—20% (Ghana) to 57%

### Strengths and limitations of this study

► This study is from one of the largest quality improvement interventions in low/middle-income countries as it harmonised the methods and developed common tools/instruments to assess respectful maternity care in 43 health facilities across the three countries.

► The study employed a variety of approaches including direct observations, interviews and records review, and triangulated the findings to comprehensively assess respectful maternity care provided at the health facilities both in the past and at present.

► The cross-sectional design and the short duration of contact with facilities only allowed for limited aspects of respectful care to be assessed but was a pragmatic consideration in order not to unduly interrupt care provision.

► Women's reported experiences and providers' reported practices may introduce biases; observing provider–client interactions may likely find improved quality of care than what is routinely provided to clientele (Hawthorne effect) and therefore, inadequacies in quality that were identified during the observations may under-represent the true extent of suboptimal provision of respectful care.

► The findings may have limited external validity due to UNICEF's equity focus in selecting areas of work in the respective countries; most, but not all, of the assessed facilities were in the most underserved districts and this may represent a selection bias in the context of the country.

(Bangladesh)—will not return to the same facilities for future childbirth.

**Conclusions** Facilities in Bangladesh, Ghana and Tanzania have foundational systems that facilitate RMC. Structural inadequacies and policy gaps pose challenges. Many women were, however, unwilling to return to the same facilities for future deliveries although they (and providers) highly rated these facilities.

## INTRODUCTION

In 2015, the WHO issued a statement calling for global actions to prevent and eliminate disrespect and abuse during facility-based childbirth and emphasised women's right to dignified and respectful care throughout pregnancy and childbirth.[1] Facility birth rates have increased due to increased awareness, demand and access to maternity services.[2] Approximately 6.5 million more women delivered with skilled providers in 2015 and, across 60 Countdown to 2030 countries, the median skilled attendance rate increased from 65% in 2015 to 77% in 2017.[3] However, optimism around the increasing trend in facility deliveries must be cautious because deaths among pregnant women and their babies (born and unborn) have declined at a much slower pace, confirming that, unless coupled with care of the requisite quality, outcomes for women and their babies may be poor[4–8] and communities may lose trust in facilities.[9 10]

D'Oliveira et al[11] and other authors found that pregnant women were abused within health facilities during childbirth; the care they receive is not respectful or dignified[11–13] and does not place any value on their expectations. The WHO framework of 2016 therefore defined eight standards for improving the quality of maternal and newborn healthcare (MNH) in health facilities along two domains—provision and experience of care.[14] It recognises the need to amplify the voices of women from low/middle-income countries (LMICs) in the design, content and delivery of the care they receive around childbirth to ensure that it respects their dignity, preferences and aspirations.[15] The eight WHO standards were not systematically tested in any LMIC. Moreover, it is not fully described how respect and dignity of care can be objectively assessed. Tools such as the Mothers on Respect Index have not been tested in LMICs.[16]

As part of a Mother and Baby Friendly Hospital Initiative, UNICEF and partners in the Ministries of Health of Bangladesh, Ghana and Tanzania are working within selected districts to effectively improve breastfeeding and MNH outcomes. The partnership used this platform to implement a quasi-experimental, pre–post study to test the feasibility of implementing the WHO standards within these LMICs and to answer these critical questions: how pervasive is disrespectful and abusive care within health facilities in LMICs? Do facilities have and enforce conducive policy environments and/or guidelines that enable provision and experience of respectful maternity care (RMC) and what are (and can we rely on) women's perspectives on the dignity around the care they experience?

Our objective was to assess RMC from provider perspectives and as experienced by clients in labour wards within the three countries. We also aimed to assess the implications of these on women's satisfaction with the care. The manuscript therefore presents findings from a baseline assessment describing existing provisions, health worker delivery and families' perceptions of RMC prior to the implementation of the standards. The results will contribute to the continuing global discussions around how to effectively assess respect and dignity in the provision and experience of maternity care.

## METHODS

### Study design and setting

This is a cross-sectional study, nested within the quasi-experimental pre–post evaluation of the Every Mother Every Newborn Quality Improvement (EMEN-QI) intervention. The protocol for the evaluation of the EMEN-QI intervention has been published.[17] This study had four components: inventory to check the availability of clean toilet and handwashing facilities, policies, protocols and guidelines for providing RMC; questionnaire and qualitative interviews with facility staff and maternity clients; structured observation of maternity care provided to clients and review of client medical records for documentation of RMC provided in the past.

It was carried out by three research institutions from the respective countries—International Centre for Diarrhoeal Diseases Research in Bangladesh, Navrongo Health Research Centre, Ghana Health Service and the National Institute for Medical Research, Tanzania. It was conducted between May and August 2016 in 43 health facilities: 15 in Bangladesh, 16 in Ghana and 12 in Tanzania, comprising 1 regional and 12 district hospitals and 30 health centres (HCs)/upazilla health complexes (UHCs). Teams of two to four clinician research assistants (CRAs) and research social scientists/anthropologists (RSAs) carried out the assessment in each facility. These teams were trained by the principal investigators (PIs) in each country under the direct supervision of the lead author. The training was followed by pretesting exercises which included hospital visits until there was consensus on the observations made for client–provider interactions.

In Bangladesh, 3 hospitals and 12 UHCs in Kurigram, Lalmonirhat and Gaibandha districts of the Rangpur division were involved. In Ghana, the regional hospital of the Upper East Region, 5 district hospitals and 10 health centres in 8 districts (Bawku, Kassena-Nankana East and Bolgatanga Municipalities and Bawku West, Bongo, Kassena-Nankana West and Builsa North districts) were covered. For Tanzania, the assessments were done in four hospitals and eight HCs within Njombe Town Council, Makete, Ludewa and Wanging'ombe districts of the Njombe region. These facilities provide basic/comprehensive emergency obstetric care (CEmOC) services and were selected in consultation with the ministries of health of the respective countries to reflect high average caseload, relatively poor MNH indices and nationally representative human resource availability.[17] All district hospitals, UHCs in Bangladesh and one HC in Tanzania provided CEmOC services.

### Participants

To cover provision and experience of care, respondents for the study included facility managers and 'in-charges',

providers who directly cared for women in labour for the 2 weeks of the assessment, recently delivered women and their birth companions. The health records of all women, who were seen and cared for at the maternity in the 3–6 months prior to the assessment, were reviewed and data abstracted.

## Informed consent

We consented all participants to participate in the study. Research assistants read out or gave an information sheet to all participants explaining the purpose of the assessment in a language of their choice. Each participant was allowed to ask for clarifications on the study. They were informed that there will be no direct benefits for participation but the information they provide will help improve the quality of maternity care provided in facilities. They were assured of their right to withdraw participation at any time without affecting their position in the facility or care they received, and that information provided will be treated as confidential. Agreement to participate was indicated with a signature/thumbprint. No respondent refused participation in the study. All interviews lasted 45–60 min.

## Patient and public involvement

The study tools and instruments were pretested in all the three countries to solicit facility-user input into the strategy for data collection. The data collection involved patients or maternity clients who have been discharged from the facilities as well as their companions who accompanied them to the facility for the delivery. Their identities were protected, and their confidentiality and privacy were ensured for all the information they provided. Their contributions have been cited in the acknowledgements.

## Assessment of RMC

RMC was assessed along the dimensions of the three components in the WHO framework—emotional support, respectful communication and respectful and dignified care—from the perspectives of the provider, the women (clients) and independent-informed assessors (CRAs). These were supplemented with data on the availability of the right policy environment and facility provisions (eg, handwashing) for RMC.

After facility entry, CRAs interviewed facility superintendents/'in-charges' and assessed whether the facility had specific protocols and guidelines for ensuring RMC including training schedules for staff at maternity units. Using a semi-structured questionnaire, the CRAs assessed whether facilities had systems for identifying abuses of pregnant women seeking or receiving care including 'whistleblower' policies where care providers could report client abuses perpetrated by their peers under protected or concealed identities. Respondents were asked how they involved communities in defining and providing quality and woman-centred maternity care. We considered the availability of clean client toilets on the maternity as a basic necessity in providing RMC and

so CRAs checked availability of maternity ward toilets and rated their cleanliness based on agreed criteria.

CRAs also interviewed care providers in the maternity wards on any formal RMC training they received, any encounter with situations of client abuse in facilities and management responses to abuses. Care providers included specialists, doctors, midwives and staff nurses. Selection was biased towards staff who play key leadership or decision-making roles in facilities (eg, matrons who provide care and enforced guidelines and regulations) or served for more than 6 months.

The CRAs also passively observed client–provider interactions as women navigated through all the points of care provision within facilities—from reception through until discharge after delivery. These provided first-hand experiences of auditory and visual privacy and provider–client communication such as explanation of the care plan and the involvement of clients in their own care. They checked whether providers ensured a conducive atmosphere for women and their companions to ask questions on the care they received; support (emotional and other) for women during childbirth and any abuses. They reviewed women's records to assess documentation of care.

RSAs identified isolated, comfortable and conducive spaces within facility to interview recently-delivered women, who had been discharged from facilities, on their experiences of the maternity care provided. All women who had been in the facility for at least 6 hours and consented to participate were interviewed irrespective of birth outcomes. This was because women who presented in labour were observed for at least 6 hours after the birth before discharge, creating opportunity for them to interact with providers. The RSAs ensured that respondents were not seen or heard during the interview. The interview covered women's perspectives on provider communication, responsiveness, respect for their choices and preferences, visual and auditory privacy during care provision, demand for informal or formal payments and withholding of care if such demands were refused, as well as any physical, sexual or verbal abuses during women's stay in the facility. Clients' impressions and satisfaction with the maternity care they experienced were also elicited but aside from that, they were asked whether their experiences will make them recommend the facility to relatives or friends for maternity care or whether they themselves will like to return to the same facility for their next childbirth.

RSAs also interviewed any persons accompanying the women during their stay in the facility for their independent perspectives on the care provided to their relative or friend. RSAs asked whether providers demanded or collected payments for services or whether any service was withheld for any reason including inability to pay.

## Quality assurance and data processing

Stringent uniform quality assurance processes were implemented starting from the field through to the data processing units of the research institutions. Team

coordinators liaised between fieldworkers, facility leadership and data centres. Data collection was paper-based in Ghana and Bangladesh but electronic in Tanzania. At the end of each day, research teams met and peer reviewed each other's forms for completeness. Team coordinators rechecked all forms to ensure blanks or inconsistencies were identified and resolved. All checked forms were batched, logged into batch control sheets and transmitted to computer centres for processing.

In Ghana and Bangladesh, two data entry clerks independently entered the data and the two data streams were compared for agreement. Any disagreements were resolved with the source questionnaire by data managers who then run range and consistency and data integrity checks to generate exception reports for the study investigators to review and resolve. In Tanzania, data were directly transmitted to central servers for review by data managers. Where required, forms with challenges were sent back to the field for resolution. Cleaned data were securely stored on password-protected servers. A copy of the final data was shared with UNICEF Headquarters.

## Data analyses

Data were transferred to Stata V.14.1 for analyses. Data were represented with tabular, numerical and graphical methods. Proportions, means (SD) and medians (IQR) were estimated for the outcome analyses. Z-tests were performed to assess differences in proportions and $\chi^2$ tests for association between variables. All tests were significant at the 5% level and 95% CIs were constructed around point estimates where necessary. Responses to open-ended questions were coded into themes in NVivo V.12.0 after repeated readings using the framework approach. Qualitative analyses involved exploration of relationships between themes and were triangulated to the close-ended questions to facilitate their contextual interpretations.

## Coordination

UNICEF coordinated the uniform assessment across countries. A consultant (AM) facilitated a UNICEF-organised design meeting and training in New York with PIs. Skype conference calls were organised every week to agree with harmonised timelines for implementation and update on progress. The coordinated approach allowed for common solutions to challenges encountered and for the consultant to visit each country during the training and initiation of data collection.

## RESULTS

Health facility managers in all 43 facilities across the three countries were interviewed. There were 325 care provider interviews with nurses, midwives and doctors comprising 158 in Bangladesh, 86 in Ghana and 81 in Tanzania. Questionnaires were administered by social scientists/anthropologists to collect data from a total of 849 recently delivered women: 295 from Bangladesh, 381 from Ghana and 173 from Tanzania. Observation of 641

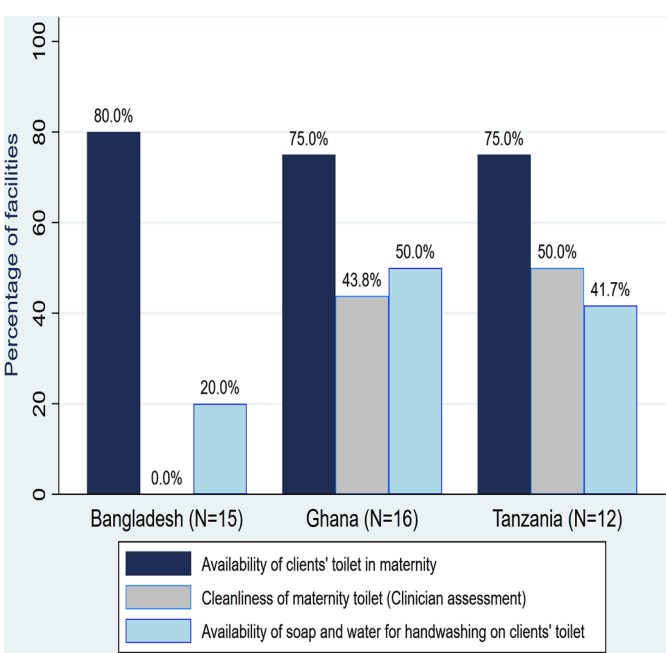

**Figure 1** Availability of client toilets with soap and water for handwashing and its cleanliness.

client–provider interactions was conducted by CRAs—387 in Bangladesh, 134 in Ghana and 120 in Tanzania.

### Availability of dedicated clean maternity toilet as part of RMC

Clean toilets with handwashing facilities were inadequate in the maternity units in all three countries (figure 1). Three (20%) facilities in Bangladesh, four (25%) in Ghana and three (25%) in Tanzania did not have client toilets in the maternity. Where they existed, they were judged as unclean by study clinicians—6 (50%) in Tanzania, 9 (56%) in Ghana and 15 (100%) in Bangladesh; and without soap and water for handwashing—7 (58%) in Tanzania, 8 (50%) in Ghana and 12 (80%) in Bangladesh. Women's perceptions on the cleanliness of the toilets contrasted with these clinician observations. Figure 2 shows that only 104 (35%) women interviewed in Bangladesh and 301 (79%) in Ghana thought the toilets were unclean.

### Availability of the right policy environment for RMC

Majority (13 representing 81.3%) of facility managers in Ghana, compared with 11 (73.3%) in Bangladesh and 6 (50.0%) in Tanzania, reported that clear processes existed for identifying client abuses (table 1). In Ghana, all but one facility had a 'whistleblower' policy for reporting abuses, and all facilities (100%) actively encourage women to report any provider abuses. In Tanzania, only one in four facilities had a 'whistleblower' policy, and one in three encouraged women to report abuses. Data on whistleblower policies were not captured in Bangladesh although, in qualitative interviews, staff suggested that mechanisms for making anonymous complaints using mobile short message service (SMS) existed, were captured onto the district health information systems platform and followed up regularly.

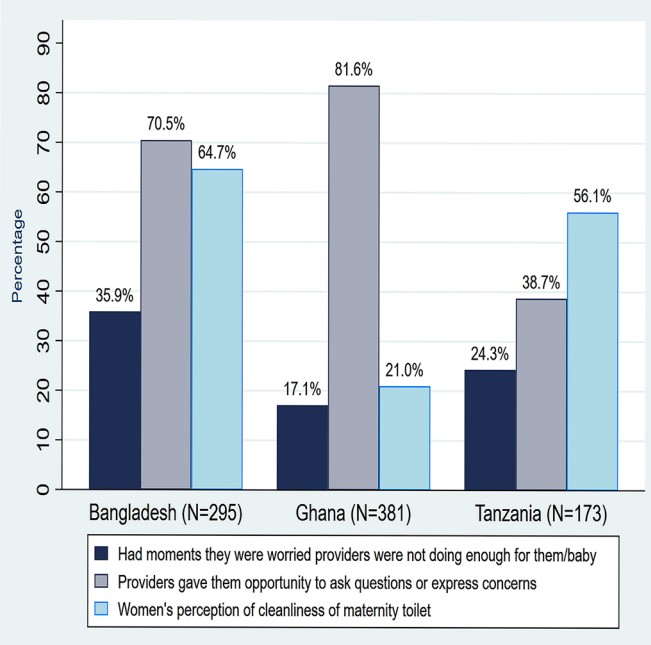

**Figure 2** Women's reported experience of care at maternity units.

Consistent with facility managers' reports, assessors found that, of all three countries, most health facilities in Ghana had policies on women's rights during care at the facility. Sixty-one (71%) providers in Ghanaian facilities reported that they had policies on addressing client concerns and 70 (82%) reported that policies existed on the rights of clients who sought maternity care (table 2). In Bangladesh, only 54 (41%) of providers interviewed reported the existence of policies to address client

concerns and 83 (53%) reported policies existed on women's rights, although the staff could not show copies of the policy documents to the assessors during the visits. In Tanzania, 54 (67%) providers reported that their facilities had policies to address client concerns, while 57 (70%) reported that policies existed on women's rights while receiving care (table 2).

### Training of providers in RMC

Specific training on how to provide maternity care with dignity and respect was not provided to 30%–60% of providers across the three countries. In Bangladesh, 127 (80%) providers could not mention two measures being implemented in their facilities to prevent abuses of women during maternity care (table 2). There were 32 (37%) providers in Ghana and 24 (30%) in Tanzania who could also not mention two or more measures. A substantial proportion—69 (80%) in Ghana and 142 (90%) in Bangladesh—did not know of two or more measures in place to treat or rehabilitate victims of abuse. Meanwhile, table 2 also shows that almost all of these providers highly rated the respect and dignity in the care they provided, scoring it above 3 on a scale of 1–5, 5 being the best—151 (96%) of them in Bangladesh, 86 (100%) in Ghana and 77 (95%) in Tanzania.

### Women's reported experience of maternity care and how it compares with assessors' observations

While most women reported that they were happy with the attitude of providers (table 3), they also said providers were not doing enough for them and/or their babies and were not given the opportunity to express these concerns (figure 2). For example, in Tanzania, although 168

| Table 1 | Facility managers' report on the availability of guidelines for identifying, responding and preventing abuses and community involvement in ensuring quality of care | | |
|---|---|---|---|
| | **Number of responses (%)** | | |
| **Guideline or policy parameter in health facilities** | **Bangladesh N=15** | **Ghana N=16** | **Tanzania N=12** |
| A process for identifying abuses of client by provider | 11 (73.3) | 13 (81.3) | 6 (50.0) |
| 'Whistleblower' policy for peer reporting under cover | Not recoded | 15 (93.8) | 3 (25.0) |
| Women are encouraged to report provider abuses | 9 (60.0) | 16 (100.0) | 4 (33.3) |
| Response to most recent abuse case (number of cases) | 1 case | 5 cases | 1 case |
| - Investigated and penal actions taken internally | 0 (0.0) | 4 (80.0) | 1 (100.0) |
| - Investigated within facility but referred for action | 0 (0.0) | 1 (20.0) | 0 (0.0) |
| - Referred to external bodies for investigation and sanction | 1 (100.0) | 0 (0.0) | 0 (0.0) |
| Measures taken to prevent future abuses† | | | |
| - Counselling and verbal warning | 1 (100.0)* | 5 (100.0) | 1 (100.0) |
| Involves community stakeholders in quality improvement efforts | 8 (53.3) | 12 (75.0) | 6 (50.0) |
| Women's support groups exist in the catchment population | 5 (33.3) | 10 (62.5) | 2 (16.7) |

*In Bangladesh, they raise awareness and supervise the staff who perpetrated the act.
†The assessment checked for other penal measures such as demotion of staff, reassignment/transfer, embargo on salary and interdiction but there was not reported in any of the facilities.

**Table 2** Care providers' report on availability of guidelines for respectful maternal and newborn care

| Subject area | Providers' account on modalities for ensuring respectful maternity and newborn care | Number of responses (%) | | |
| --- | --- | --- | --- | --- |
| | | Bangladesh N=158 | Ghana N=86 | Tanzania N=81 |
| Policies on women's rights | Policy and procedure for addressing patient concerns | 54 (41.1) | 61 (71.0) | 54 (66.7) |
| | Policies on the rights of patients | 83 (52.5) | 70 (81.6) | 57 (70.4) |
| Staff training on respectful care | Staff trained on how to treat childbearing women with respect | 64 (40.5) | 60 (70.1) | 61 (75.0) |
| Staff rating of RMC provided in their facility | Staff who rated their facility above 3 on a scale of 1–5 (5 being best) for care with respect and dignity they provide to women | 151 (95.6) | 86 (100.0) | 77 (95.1) |
| Identification and response to abuse | Process exist for identifying and reporting abuses during maternity and newborn care | 46 (29.1) | 58 (67.8) | 41 (50.0) |
| | Have encountered abuse victims in their maternity and newborn care practice | 42 (26.6) | 28 (32.2) | 16 (19.8) |
| | Staff who knew of 2 or more measures being implemented in their facilities to prevent abuses | 31 (19.6) | 54 (62.7) | 57 (70.4) |
| | Staff who mentioned 2 or more measures being implemented in their facilities to treat or rehabilitate victims of abuses | 16 (10.1) | 17 (19.5) | 56 (69.0) |
| | Community stakeholders' involvement in addressing disrespect and abuse of women during childbirth | 49 (31.0) | 49 (57.5) | 27 (33.3) |
| Payment for services | Women are expected to pay for normal deliveries | 32 (20.3) | 7 (8.1) | 42 (51.9) |
| | Women are expected to buy supplies for normal deliveries | 132 (83.5) | 73 (85.1) | 61 (75.0) |
| | Women are expected to pay before treatment in case of obstetric emergencies | 19 (12.0) | 9 (10.3) | 27 (33.3) |

RMC, respectful maternity care.

(97.1%) of women reported that providers treated them with 'respect' (table 3) and 161 (93.1%) said providers were responsive to them when they asked for support, figure 2 also shows that about a quarter of them, 42 (24.3%) reported moments when they were worried that the providers were not doing enough for them or their

**Table 3** Women's report of abuses, provider attitudes and overall satisfaction with care provided

| Aspect of care | Modality for which women's perspective around facility experiences was assessed | Number of responses (%) | | |
| --- | --- | --- | --- | --- |
| | | Bangladesh N=295 | Ghana N=381 | Tanzania N=173 |
| Abuses during caregiving | Physical abuse from providers at the facility | 6 (2.0) | 9 (2.4) | 2 (1.2) |
| | Verbal abuse from providers at the facility | 19 (6.4) | 25 (6.6) | 8 (4.6) |
| | Sexual abuse from providers at the facility | 2 (0.6) | 2 (0.7) | 1 (0.6) |
| Provider relationship with caregivers | Their concerns were considered by providers in caring for them | 224 (75.9) | 352 (92.4) | 108 (62.4) |
| | Providers were responsive when asked for support | 262 (88.8) | 366 (96.1) | 161 (93.1) |
| | They were treated with respect by providers | 266 (90.2) | 367 (96.1) | 168 (97.1) |
| Satisfaction with care | Satisfied with attitude of providers | 271 (91.7) | 353 (92.7) | 161 (93.1) |
| | Satisfied with information received from providers on breast feeding | 191 (64.8) | 338 (88.7%) | 132 (76.3) |
| | Satisfied with information received from providers on family planning | 38 (12.9) | 208 (54.6) | 77 (44.5) |

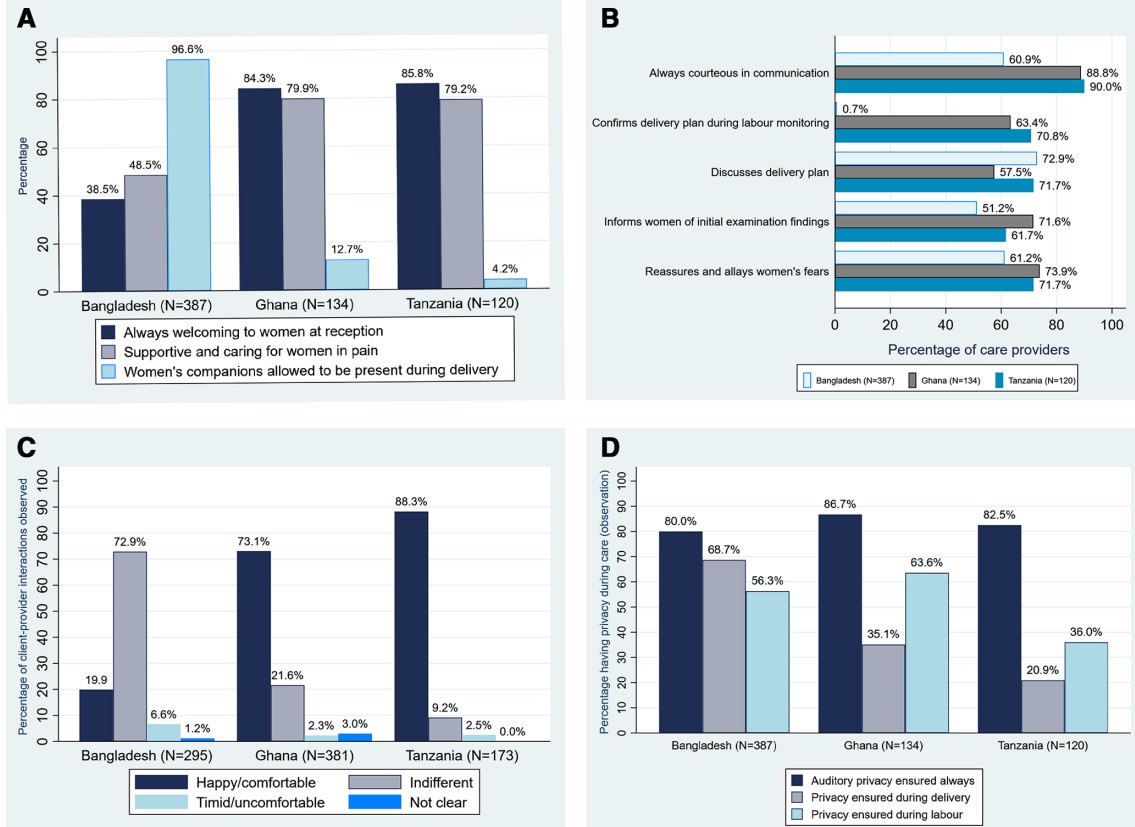

**Figure 3** (A–D) Assessors' observations on women's maternity care experiences during provider interactions in health facilities.

babies. Moreover, 106 (61.3%) reported that they had no opportunity to express these concerns or ask questions. The trends were similar in Ghana and Bangladesh; in Bangladesh, for instance, 266 (90.2%) women reported being treated with 'respect' but 106 (35.9%) had moments they were concerned that the providers were not doing enough for them and 87 (29.5%) thought they were not given the opportunity to express these concerns.

During the observation of client–provider interactions by the study assessors, varying trends were found between the three countries. Figure 3A,B shows that in Ghana and Tanzania, at reception, the providers were mostly welcoming—113 (84%) and 103 (86%), respectively—and were supportive and caring when women were in labour pain, 107 (80%) and 95 (79%), respectively. The assessors also found, on a Likert scale between 'never', 'only sometimes', 'mostly' to 'always', providers were 'always courteous' in communication with the women—119 (89%) in Ghana and 108 (90%) in Tanzania. In the two countries, providers informed 62%–72% of women about their initial examination findings, discussed delivery plans with 52%–72% of them, reassured them and allayed their fears in over 70% of cases, and confirmed delivery plans during labour monitoring in 63%–71% of cases. However, figure 3C shows that in these same countries, 12%–27% of respondents were not 'comfortable' talking to the providers and 2.3%–2.5% were visibly uncomfortable or looked timid.

Only 3% of women in the two countries made specific requests from the providers and none in Ghana could request for a birth companion (table 4).

In contrast, the findings were different in Bangladesh as uptake of these modalities in RMC was lower. Figure 3A,B shows that providers were 'always welcoming' to 149 (36%) of women, were supportive and caring during labour pains in 188 (49%) and just about half, 198, (51%) were told examination findings at initial examinations. Though the delivery plan was discussed with 282 (73%) of the women, during labour monitoring, these plans were confirmed with only 3 (0.7%) of them. Table 4 shows that about 14 (4%) of women in Bangladesh made specific request from the providers and though all were refused, it was only in a single case that the refusal was done respectfully. Figure 3C shows that 25 (7%) of women in Bangladesh felt uncomfortable and timid when talking to the providers, about three times those in Ghana and Tanzania.

**Companion at birth**

No woman in Ghana and only two (2%) in Tanzania reported requesting a birth companion for the delivery (table 4). This contrasts with the 286 (74%) in Bangladesh. However, in figure 3A, women's companions were allowed to be present for 17 (13%) of women in Ghana, 5 (4%) in Tanzania and 374 (97%) in Bangladesh.

**Table 4** Assessor observations of the provider attitudes during maternity care for women

| Aspect of care | Modalities of care that were observed by assessors | Number of responses (%) | | |
| --- | --- | --- | --- | --- |
| | | Bangladesh N=387 | Ghana N=134 | Tanzania N=120 |
| Client requests | Women who made request for anything | 14 (3.7) | 4 (3.0) | 4 (3.3) |
| | - Those whose requests were refused with respect | 1 (6.7) | 4 (100.0) | 3 (75.0) |
| | Women who requested for a birth companion | 286 (73.9) | 0 (0.0) | 2 (1.7) |
| Payments at facility and effect on care | Women/families paid money for some services | 164 (42.4) | 19 (14.2) | 61 (50.8) |
| | Women/families who complained about the payment | 45 (11.6) | 1 (0.8) | 22 (18.3) |
| | Women for whom services were withheld due to inability to pay | 2 (0.5) | 0 (0.0) | 13 (10.8) |

## Verbal, physical or sexual abuse of women during maternity care

Facility managers' interviews (table 1) showed that cases of abuse of women during maternity care were reported but measures to prevent recurrence were weak. Managers were aware of recent (less than 2 weeks) cases of abuse in their facilities. Five cases of abuse were reported in Ghana compared with only one each in Bangladesh and Tanzania. Staff also reported encounters with victims of abuse in their practice. About 16 (20%) providers in Tanzania, 42 (27%) in Bangladesh and 27 (32%) in Ghana had encountered victims of abuse in their practice but up to 70% of providers between the three countries did not personally know of any existing systematic processes for identifying and reporting abuses within their facilities, although managers reported that these processes existed. Meanwhile, women reported that they were physically, verbally or sexually abused by providers during care (table 3). The most common form of abuse was verbal; on average, 6% of women in all three countries experienced this form of abuse (5% in Tanzania, 6% in Bangladesh and 7% in Ghana). Sexual abuses were the rarest with only one or two women reporting the experience in each country. On average, 2% of the women also experienced physical abuses from providers during their stay in the facility.

When these abuses occurred, facilities in Ghana and Tanzania conducted internal investigations and the only action taken to prevent future occurrences was counselling or verbal warnings to the staff involved (five of the six cases). In comparison, the only recent case in Bangladesh was referred to an external body for investigation and action.

## Privacy of care

During assessors' observations of client–provider interactions, of the three countries, auditory privacy was ensured in over 80% of interactions (figure 3D). Privacy during labour was ensured mostly in Ghana where there was privacy in 64% of the 134 client–provider interactions observed. In Bangladesh, privacy in labour was ensured for 218 (56%) of interactions and it was for 43 (36%) in

Tanzania. During delivery care, privacy was least ensured in Tanzania (21% of 120 observations). It was 35% in Ghana and 69% in Bangladesh.

## Demand for payment and withholding of delivery care services for inability to pay

Consistent with findings from CRA observations of client–provider interactions, providers in all three countries reported that women were made to pay money for supplies and/or delivery, even in obstetric emergencies (table 2). Charging fees was most common within facilities in Tanzania where 42 (52%) providers reported that women were made to pay fees for normal deliveries. Twenty-seven (33%) providers in Tanzania said these payments were also expected for obstetric emergencies. The proportion of providers reporting these payments was relatively lower in Bangladesh where 32 (20%) reported payments for normal deliveries and 19 (12%) in emergencies. In Ghana, where a National Health Insurance Scheme covers the cost of facility births, seven (8%) providers reported that some illegal fees were still charged for normal deliveries and nine (10%) said there were fees charged for obstetric emergencies. Seventy-three (85%) providers in Ghana, 132 (83%) in Bangladesh to 61 (75%) in Tanzania reported that women had to buy supplies for normal deliveries in the facility.

During observation of client–provider interactions, study assessors found that providers demanded money from women for maternity services. This practice was most common in Tanzania where the demands were made for 61 (51%) cases observed, followed by 164 (42%) in Bangladesh and 19 (14%) in Ghana. Maternity services were sometimes withheld from women when they were unable to pay and, only a few women complained about such payments. During observations, 1 mother in Ghana, 45 (12%) in Bangladesh and 22 (18%) in Tanzania were seen complaining about the charges. It is noteworthy that some delivery care services were withheld from women (11% in Tanzania and 0.5% in Bangladesh) because of the inability to pay. No such service was withheld in Ghana.

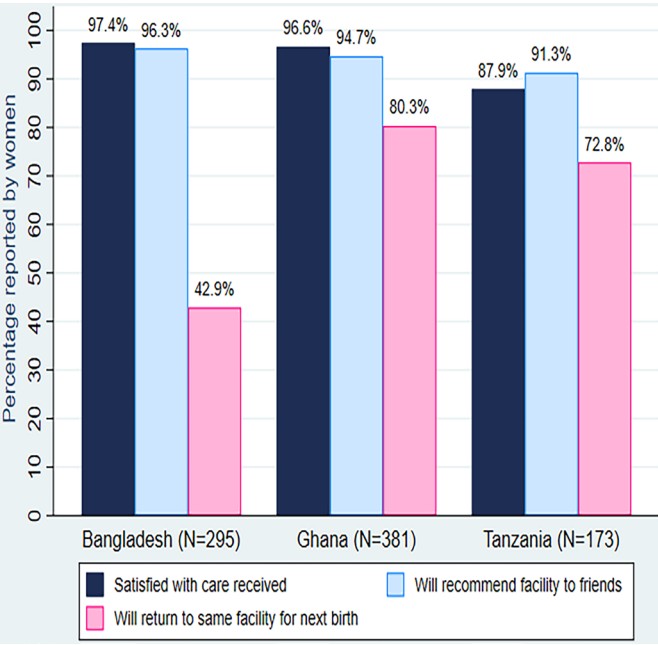

**Figure 4** Mothers' satisfaction with care received in the health facilities during childbirth in the three countries.

## Community involvement providing RMC

Managers of facilities reported that 6 (50%) facilities in Tanzania, 8 (53%) in Bangladesh and 12 (75%) in Ghana involved community stakeholders in deliberations around ensuring RMC (table 2). Facility providers corroborated this. When care was abusive or not respectful, 49 (31%) providers in Bangladesh, 27 (33%) in Tanzania and 49 (58%) in Ghana reported that their facility involved community stakeholders in addressing it.

## Overall satisfaction with the maternity care at health facilities

On a scale of 1–5 (5 being the best) in terms of RMC provided to women, all (100%) providers from Ghana, 151 (96%) from Bangladesh and 77 (95%) from Tanzania ranked their facilities 3 and above. Over 90% of women from all countries also reported satisfaction with the attitude of the providers—271 (91.7%) in Bangladesh, 353 (92.7%) in Ghana and 161 (93.1%) in Tanzania (table 3). Figure 4, however, shows that majority of women expressed satisfaction with the maternity care they received at facilities—287 (97%) in Bangladesh but, on the contrary, only 106 (43%) will return to the same facilities for future maternity care and this difference was statistically significant (p<0.001). Similarly, 386 (97%) in Ghana expressed satisfaction with the care but only 305 (80%) will return to the same facility for the next delivery; while in Tanzania, 152 (88%) were satisfied but only 126 (73%) will return to the same facilities in the future for maternity care showing differences that were statistically significant (p<0.0001). Meanwhile, these same women would recommend the facility to a relative or friend for maternity care even if they themselves will not do the same in the future—96% in Bangladesh, 95% in Ghana and 91% in Tanzania.

## DISCUSSION

Though RMC is a relatively new and developing concept in many LMICs, our assessment in the three countries found that the infrastructure, policies on caregiving and staff readiness for providing RMC were in place to build on. Prior to implementing EMEN-QI, uptake of practices around the three RMC domains (effective communication, supportive care and dignified and respectful care) was variable across the three countries but generally high. Three-quarters and more of the facilities had separate toilets in the maternity for women and half of these had handwashing facilities. Providers' and managers' reports showed that approximately 70% facilities in Ghana and Tanzania had policies on patients' rights and procedures for addressing their concerns. Only in 40%–50% in Bangladesh had the same and without this right policy environment, women in Bangladesh did not have good reception, examination findings were not communicated to them, their concerns were not considered in the care they received and they were not supported during labour. Structural inadequacies did not allow for privacy during maternity care for women and birth companions were also not present especially in Tanzania and Ghana. Moreover, illegal payments were demanded from women for delivery care and services were sometimes withheld due to inability to pay. With the right processes for ensuring patient rights, including 'whistleblower' policy as in Ghana, abuse reporting was higher. Facility providers over-rated RMC they provided as compared to what was observed. Between 50% and 75% of facilities already involve community stakeholders in addressing RMC experiences of women. Although most women across the three countries reported being 'satisfied' with care provided at facilities, they will not come back to the same facilities for their next delivery based on their experiences.

Our findings align with those from earlier studies suggesting that assessing care satisfaction with questions that merely ask whether women or providers were satisfied with care received or provided was unreliable.[9] Our results show that while 88%–97% of women and providers highly rated their satisfaction with care received or provided, perhaps a probable indicator of satisfaction with care may be the significant 20%–58% (p<0.0001) of women who reported that, based on the maternity care experienced, they will not return to the same facilities for birth in the future. This apparent inconsistency between women's reported satisfaction with care and desire to return to the same facility has not often been assessed although alluded to when researchers find unbelievably high reported level of satisfaction under situations that did not appear to merit the same.[9 10] Lack of respectful and dignified care is a known barrier to subsequent care-seeking.[18] This indicator will require testing in robust studies.

There appeared differences in observed provider attitudes between the South Asian and sub-Saharan African facilities in that, apart from Bangladesh, figure 3A shows that providers were mostly welcoming to women and

supported them during labour pains. With only 39% of providers welcoming women and 48% supporting in Bangladesh compared with over 80% in Ghana or Tanzania, it was not surprising that about 7% of women felt intimidated to talk to providers. Meanwhile, 17%–36% of women across the three countries were sometimes concerned that the providers were not providing enough care for them but could not express their feelings. The UK quality care commission[19] defines respect and dignity in care as care 'that treats care recipients as equals' who merely require support from care providers to be autonomous and independent.[20] This definition conforms with the fourth, fifth and sixth WHO standards for facility quality of care for mothers and newborns.[14]

Abuse is seen as a way of controlling clients to undermine their autonomy.[11] The assessment findings confirm that across the three countries, physical, sexual, verbal and psychological abuses (where services are withheld from women because of inability to pay informal/illegal charges) were experienced by women. These findings concur with reports of care providers 'scolding' or 'shouting at' women and verbalising words that were 'unkind, brusque, rude, unsympathetic and uncaring' to them.[21 22] We believe that the 4.6%–6.6% of verbal abuses may represent gross under-reporting and that the challenge might be bigger. Also, providing RMC requires development of infrastructure. For instance, none of the three countries prohibits the use of birth companions but the set-up of the units did not allow for privacy or birth companions in these units.

Respectful care is also not merely the absence of abuses. Policies on the rights of maternity clients and for addressing abuse are not always present in facilities particularly in Tanzania. Even when they existed, providers were oblivious, or the enforcement was weak. A conducive policy atmosphere facilitates reporting of abuse cases. Our finding that 25% (Tanzania) to 60% (Bangladesh) of providers have not been trained on provision of dignified and RMC is unfortunate. For instance, Ghana reported the most advanced policy environment to identify and address abuses including a 'whistleblower' policy. Consequently, women in Ghana were the most likely to report of abuses. This suggests that addressing abuses in maternity care requires deliberate efforts to create conducive environments that address system rather than individual failures (that perpetrate a 'blame' culture)—literally 'a systems approach in a person-centred system'.[23 24]

The WHO statement affirms every woman's right to dignified, respectful healthcare, identifying an increasing need for more research to measure the burden and pervasiveness of disrespect and abuse in health facilities.[1 21 22 25] Vogel et al[25] added that poor measurement of respectful care may result from the lack of globally applicable terminologies.

Initiatives to improve respect around childbirth care are not new in these countries although on smaller scales: Bangladesh has implemented a Women Friendly Health Services Initiative in some facilities but Kurigram district did not benefit. Similarly, the Ifakara Research Institute in Tanzania is also implementing RMC interventions in some facilities. These have not permeated the healthcare systems. That quality affects outcomes is incontrovertible.[26] We believe that efforts to provide RMC must be intentional, goal directed, measurable, backed by a strong political will, and should have robust systems of measurement and accountability.[27]

This study has many strengths: first, using harmonised methods to assess the 43 facilities makes it one of the largest multicountry facility assessments from which data are amenable to direct cross-country comparisons. Second, the use of a variety of approaches including observations and data review increased the objectivity and improved understanding of how prevailing practices measure up to the standards at individual, structural and policy levels according to Freedman et al's framework.[28]

We had limitations: the cross-sectional design does not allow for attribution of causality and the short duration of contact with facilities limited the aspects of respectful care that was assessed. However, it was the most pragmatic design for many settings if it should not unduly interrupt caregiving. Women's reported experiences and caregiver's practices were subject to biases. Observing provider–client interactions may likely to make them go beyond what they usually do (Hawthorne effect). Consequently, inadequacies identified during the observations may represent a small dip in an abyss of inadequate respect in care. These findings may not be generalisable to all facilities in the respective countries because, due to UNICEF's equity focus, assessed facilities were located in most under-served and least developed districts which may imply selection biases. The assessment did not measure the psychometric properties of any of the indicators and merely generated proposed candidate indicators for future testing.

This study contributes to amplifying women's voices, which is key to the values of the WHO/UNICEF-coordinated Quality, Equity, Dignity network and the recent petition on 'What women want?'[29] However, women's aversion to care that is not dignified or respectful, nuanced in their decision not to patronise these facilities in the future should alert health systems that community trust in facilities for delivery care should not be taken for granted. The Lancet Quality Commission advocates for measurement of client satisfaction on a systematic basis.[9]

In conclusion, RMC provided and consequently experienced by women across the three countries had many good aspects that could form a foundation for quality improvement interventions. Health providers' accounts over-rated the respect and dignity in the care they provide to women way beyond what was directly observed. Critical gaps still exist which a systematic approach to implementation of RMC, as envisaged in the EMEN-QI initiative, should address. It calls for more rigorous measurement of efficacy of these measures as well as in-depth anthropological investigation to understand the drivers of respectful and dignified care to help develop corrective strategies

and accountability mechanisms around these in a participatory manner. As a first step, respectful care must form a critical component of preservice training in all countries. Community engagement through organising for a systematic demand creation around respectful care, follow-up and inclusion in a holistic system response will be key. Periodic client exit interviews, use of mobile SMS, comment boxes or easy-touch buttons for client feedback will be useful but need to be coupled with capacity to retrieve these data, systematically analyse them and use them to inform changes. Health systems and facilities should be bold to empower an approachable community ombudsman to listen to client complaints and provide systematic feedback into the delivery of RMC in facilities.[30] Until then, our mothers will not return to our facilities for their next childbirth after their experiences with providers in our facilities.

**Author affiliations**
[1]Epidemiology and Disease Cintrol, School of Public Health, University of Ghana, Legon, Accra, Ghana
[2]Centre for Maternal and Newborn Health, Liverpool School of Tropical Medicine, Liverpool, UK
[3]Headquarters, UNICEF, New York City, New York, USA
[4]National Institute for Medical Research, Dar es Salaam, United Republic of Tanzania
[5]Navrongo Health Research Centre, Navrongo, Ghana
[6]Maternal and Child Health Division (MCHD), International Centre for Diarrhoeal Diseases Research, Dhaka, Bangladesh

**Acknowledgements** This work was conducted under the UNICEF-BMGF MNCH partnership. We also appreciate the contributions from Dr Priscilla Wobil (UNICEF Ghana), Masum Billah (ICDDR,B Bangladesh), Stella Kilima (NIMR, Tanzania), Francis Yeji (NHRC/GHS, Ghana), management and staff of our collaborating institutions at the International Centre for Diarrhoeal Disease Research, Bangladesh, Navrongo Research Centre of the Ghana Health Service, National Institute of Medical Research, Tanzania, and Dr Tedbabe Hailegebriel, Dr Williband Zeck, Christopher Borje and Seun Oyedele of UNICEF HQ and the UNICEF country offices in the three countries. The multiple consultations with the wide range of stakeholders including global/international experts and professionals in maternal and newborn health, ministries of health and partners in Bangladesh, Ghana and Tanzania were very beneficial. We also thank the mothers who participated in the design and provided data for this analysis.

**Contributors** The study was designed by AM, NZ, EM, JW, and SEA with technical input from CB. AM, NZ, EM, JW and SEA formed a team that managed and supported the respective countries to implement the evaluation. AM and NZ drafted the manuscript and obtained input from the coauthors: JW, EM, SEA and CB. The revised manuscripts were proofread by all authors who helped in finalising it. All coauthors (NZ, AM, EM, JW and SEA) read the final manuscript and agreed to its submission for publication in this journal.

**Funding** The study was funded by the Bill and Melinda Gates Foundation through UNICEF Headquarters, New York (Grant Number OPP1112117).

**Disclaimer** The findings and conclusions in this report are those of the authors and do not represent the official position of these organisations.

**Competing interests** None declared.

**Patient consent for publication** Not required.

**Ethics approval** The study protocol was approved by institutional ethics review boards (IRBs) in all three countries: ICDDR,B in Bangladesh (PR-16024, 1st June 2016), Ghana Health Service (GHS) in Ghana (NHRCIRB226, 6th April 2016) and NIMR in Tanzania (NIMR/HQ/R.8a/Vol. IX/2176, 11th April 2016). Clearance was sought from the facility managers for the study in the respective facilities.

**Provenance and peer review** Not commissioned; externally peer reviewed.

**Data availability statement** Data may be obtained from a third party and are not publicly available. The data analysed for this manuscript are part of the bigger EMEN assessment being undertaken by UNICEF. Access to the data will follow site-specific and national guidelines of the three countries and agreements with UNICEF HQ.

**ORCID iD**
Alexander Manu http://orcid.org/0000-0001-5230-6413

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
