## [Reviewer comments · BMJ Open]

ARTICLE DETAILS

TITLE (PROVISIONAL)	Respectful Maternity Care delivered within health facilities in Bangladesh, Ghana and Tanzania - A cross-sectional assessment preceding a quality improvement intervention
AUTHORS	Manu, Alexander; Zaka, Nabila; Bianchessi, Christina; Maswanya, Edward; Williams, John; Arifeen, Shams E.

VERSION 1 – REVIEW

REVIEWER	Rose L. Molina, MD, MPH Beth Israel Deaconess Medical Center, USA
REVIEW RETURNED	12-May-2020

GENERAL COMMENTS	This revised manuscript is improved from the original submission. However, significant issues remain: General: - English grammar and syntax are improved but still many typos and errors throughout that need to be addressed.- Please use standard reporting of data throughout. There is no consistent way of reporting results throughout the manuscript. Abstract: - Please include number of facilities included- Please include most important results by country in the Results section (not ranges) Methods: - The authors do not provide a compelling reason for measuring satisfaction. This is a known measurement problem in global maternal health - satisfaction is not a meaningful construct. The authors need to better explain why they chose to collect and report on it given its known limitations.- The authors need to explain the instrument development re: measuring RMC components in much greater detail. There appears to be minimal to no psychometric properties reported, unless a validated instrument was used. More details are needed in how the data collection tools were assessed for reliability, validity.- Data analysis section requires much more information re: quantitative statistical testing and qualitative methods.- Please explain how confidentiality was preserved when participants were interviewed in facilities and their birth companions were included in the interviews. Results: - Need to summarize key findings from tables only. Also need to differentiate results by country.- It is difficult to differentiate results from quantitative survey vs. themes from qualitative interviews. This needs to be clarified
--

	throughout the results section and tables. Discussion:  - Please summarize key results in first paragraph according to standard practice. - Limitations section needs to be its own paragraph, not interspersed throughout Table 2: why dichotomize scale in row 2?
--	--

REVIEWER	Ribka Amsalu Save the Children
REVIEW RETURNED	18-May-2020

GENERAL COMMENTS	This is an interesting study and quite good data on RMC and important. My main concern is that the reporting of the study is biased towards a particular research result. The research team need to be careful not to drive the results to a particular result. I recommend rewrite of the abstract and thorough editing of the discussion section to align with the study findings. Abstract: main outcome described is “RMC provided and/or experienced by women”. I recommend to have the results and conclusion in the abstract reflect on the main outcome. Result: only one result in abstract answers that question. The other results are structural issues that are important but were not the main outcome of the study. “RMC policies and processes existed in only few facilities.” That is not in line with table 2 where majority of the HF had policies, especially in Ghana and Tanzania. “..abuse reporting was commoner(7%)” – English edit was common. Conclusion: “There are substantial gaps in RMC provided and consequently experienced by women in the three countries”. What is substantial? is <10% verbal, physical, and sexual abuse as described in table 1 and 3 considered substantial? if yes, based on what grading system? “Asking women to rate satisfaction with care experienced may be unreliable” I don’t understand how the researchers reached that conclusion. The study was a cross-sectional descriptive study and it did not rest “reliability” of source of information or “validity” of question. This conclusion is not backed with the results of the study nor the methodology used by the study. Future return could be influenced by factors not related to RMC, e.g. cost of care, distance travelled for care, etc Limitation of study “Also, women’s reported experiences and caregiver’s practices may introduce respondent biases. Observing provider-client interactions may likely find improved quality of care than what is routinely provided to clientele (Hawthorne effect). Consequently, inadequacies identified during the observations may under-represent the true extent of sub-optimal provision of respectful care.” Again, the researchers need to be careful not to drive the result in one direction. Three sources of information is a strength and not a limitation and there is no indication that the methods will “under-represent” findings. Results Page 9. Line 37 – 49. This is a descriptive data, avoid terms “ least likely” “most likely”, state findings as they are. Majority (81% and 73%), as compared to Tanzania 50%. Page 10. Line 24 -39. “women’s account...appeared to be
---

	inconsistent” a better term might be varied by question or domain of question. “women thought they were treated” a better word would be women reported that they were treated. “over 106 (61.3%)” recommend to delete over. Line 56. “none in Ghana could request for a birth companion” could not request or did not request? We should not assume that women want to have a birth companion. Page 11. Line 26-31. “...maternity care were not uncommon..” this means it was common. While Table 1 shows it was very rare – only 7 cases reported. Line 33. First sentence - Page 12. “Demand for payment and withholding of delivery care services for inability to pay” this section is confusing. Is fee for use of service and supplies the policy in each country? The tone of the section sounds that they were asked to pay out of the standard procedure. The standard procedure should be stated at the beginning of the paragraph so that the reader is informed. Page 13. “Care providers thought that the care they provided to women was with dignity and respect, but mothers and families seemed to have different perceptions on the care they experienced.” This statement is not backed by the findings of the study. Suggest to delete. The rest of the paragraph shows that both providers and women had >90% satisfaction of care provided and is consistent. I am not sure how information provided on breastfeeding and family planning relates to RMC. This is more on comprehensiveness of care provided and not so much on RMC. Unless the women asked and were not provided with the information. “ return to the same facility”. Would have been informative to see references or other studies that tested if there is association between RMC and future care preference or return to the same facility. Or not. I don’t think this study was designed to test that association. Discussion Page 13. Line 49/50. “Many women were not allowed to express opinions in the care” were not allowed or were not comfortable. These are two different dimensions requiring different response.
--	---

VERSION 1 – AUTHOR RESPONSE

Reviewer: 1

This revised manuscript is improved from the original submission. However, significant issues remain:

Response: Thank you very much for your invaluable comments.

General:

- English grammar and syntax are improved but still many typos and errors throughout that need to be addressed.
- Please use standard reporting of data throughout. There is no consistent way of reporting results throughout the manuscript.

Response: This has been done

Abstract:

- Please include number of facilities included

Response: This has been done

- Please include most important results by country in the Results section (not ranges)

Response: This has also been done as advised. Thank you.

Methods:

- The authors do not provide a compelling reason for measuring satisfaction. This is a known measurement problem in global maternal health - satisfaction is not a meaningful construct. The authors need to better explain why they chose to collect and report on it given its known limitations.

Response: This is a valid question. Our study was an evaluation of an on-going programme. It was not powered to measure psychometric properties of any indicator. We however hypothesized that, women who were satisfied, barring the challenges with distance and financial access, should be willing to return to the same facility for the next delivery care. We reported this and the manuscript has been revised to make it clear that this remains a hypothesis that will require robust testing. We used satisfaction for lack of a valid, robust and tested alternative comparator.

- The authors need to explain the instrument development re: measuring RMC components in much greater detail. There appears to be minimal to no psychometric properties reported, unless a validated instrument was used. More details are needed in how the data collection tools were assessed for reliability, validity.

Response: Thanks. Our response to the previous question addresses the issues raised here. Indeed, the assessment was built into the existing programme and was not intended to measure psychometric properties. Our earlier version made the hypothesis sound strong and we have therefore revised to clearly indicate that it is only being proposed for further testing, given its own limitations.

- Data analysis section requires much more information re: quantitative statistical testing and qualitative methods.

Response: This has been revised accordingly under "data analysis" subsection.

- Please explain how confidentiality was preserved when participants were interviewed in facilities and their birth companions were included in the interviews.

Response: This has been described in detail in the 5th paragraph under "Assessment of RMC". In summary, we selected women who have been discharged and took them to a comfortable and isolated place out of the view of the providers for the interviews. Their companions supported with the babies' care. When the woman's interview was completed, we independently interviewed the companions.

Results:

- Need to summarize key findings from tables only. Also need to differentiate results by country.

Response: This has been done.

- It is difficult to differentiate results from quantitative survey vs. themes from qualitative interviews. This needs to be clarified throughout the results section and tables.

Response: This has been done. Thank you very much.

Discussion:

- Please summarize key results in first paragraph according to standard practice.

Response: This has been done. Thank you very much.

- Limitations section needs to be its own paragraph, not interspersed throughout

Response: This has been done and it has been separated from the study strengths

Table 2: why dichotomize scale in row 2?

Response: This was to simplify the table. As can be seen from the results, those who ranked their facilities from 1 to 3 on the scale were very few. We chose 3 out of five as that is the first level that exceeds average performance.

Reviewer: 2

Reviewer Name: Ribka Amsalu

Institution and Country: Save the Children

Please state any competing interests or state 'None declared': None declared

Please leave your comments for the authors below

This is an interesting study and quite good data on RMC and important.

My main concern is that the reporting of the study is biased towards a particular research result. The research team need to be careful not to drive the results to a particular result. I recommend rewrite of the abstract and thorough editing of the discussion section to align with the study findings.

Response: We agree that inadvertently, our presentation of the results looked more at the negative than the positive findings. perhaps this was driven by the aim of the baseline assessment to identify the gaps that will inform the EMEN-QI intervention. We have therefore revised the manuscript the correct this bias.

Abstract: main outcome described is "RMC provided and/or experienced by women". I recommend to have the results and conclusion in the abstract reflect on the main outcome.

Response: This has been done. Thank you very much.

Result: only one result in abstract answers that question. The other results are structural issues that are important but were not the main outcome of the study.

Response: This has been corrected and the entire abstract re-written

"RMC policies and processes existed in only few facilities." That is not in line with table 2 where majority of the HF had policies, especially in Ghana and Tanzania.

Response: We have corrected this report to align with the findings.

"..abuse reporting was commoner(7%)" – English edit was common.

Response: This has been revised. Thank you very much.

Conclusion: "There are substantial gaps in RMC provided and consequently experienced by women

in the three countries". What is substantial? is <10% verbal, physical, and sexual abuse as described in table 1 and 3 considered substantial? if yes, based on what grading system?

Response: We agree with your concerns about the wording of the conclusion and have consequently revised it. Our principled stance for abuses was that, 7% meant that 7 out of every 100 women experienced abuses. In these facilities where several thousands of women go for delivery care, this translates to substantial numbers of women being abused. We have however changed the language to read as "critical" gaps rather than "substantial".

"Asking women to rate satisfaction with care experienced may be unreliable" I don't understand how the researchers reached that conclusion. The study was a cross-sectional descriptive study and it did not rest "reliability" of source of information or "validity" of question. This conclusion is not backed with the results of the study nor the methodology used by the study. Future return could be influenced by factors not related to RMC, e.g. cost of care, distance travelled for care, etc

Response: This is a valid point. We have revised the sentence accordingly. The point about unreliability was a quote from a previous study to which our findings align. They used unreliable as an English word as opposed to statistical reliability. We have also included in our limitations that the assessment was not aimed at measuring psychometric properties of indicators and so any suggestions made in the paper were revised to be less emphatic and more of a proposal the will require robust testing.

Limitation of study

"Also, women's reported experiences and caregiver's practices may introduce respondent biases. Observing provider-client interactions may likely find improved quality of care than what is routinely provided to clientele (Hawthorne effect). Consequently, inadequacies identified during the observations may under-represent the true extent of sub-optimal provision of respectful care." Again, the researchers need to be careful not to drive the result in one direction. Three sources of information is a strength and not a limitation and there is no indication that the methods will "under-represent" findings.

Response: Thank you for the observation. This has been revised.

Results

Page 9. Line 37 – 49. This is a descriptive data, avoid terms "least likely" "most likely", state findings as they are. Majority (81% and 73%), as compared to Tanzania 50%.

Page 10. Line 24 -39. "women's account...appeared to be inconsistent" a better term might be varied by question or domain of question. "women thought they were treated" a better word would be women reported that they were treated. "over 106 (61.3%)" recommend to delete over. Line 56. "none in Ghana could request for a birth companion" could not request or did not request? We should not assume that women want to have a birth companion.

Response: We have revised the entire results section to maintain consistency and to streamline the language.

Page 11. Line 26-31. "...maternity care were not uncommon.." this means it was common. While Table 1 shows it was very rare – only 7 cases reported. Line 33.

Response: Thank you for the observation. Medically, not uncommon does not mean common but we

agree that it may be confusing to the reader and so the language has been changed based on your recommendation.

First sentence -

Page 12. "Demand for payment and withholding of delivery care services for inability to pay" this section is confusing. Is fee for use of service and supplies the policy in each country? The tone of the section sounds that they were asked to pay out of the standard procedure. The standard procedure should be stated at the beginning of the paragraph so that the reader is informed.

Response: Thanks for this comment. We have revised the text to clearly indicate the standard procedure in the countries. What we reported were illegal charges and the supplies were not supposed to be sold in the facilities due to possible conflicts of interest.

Page 13. "Care providers thought that the care they provided to women was with dignity and respect, but mothers and families seemed to have different perceptions on the care they experienced." This statement is not backed by the findings of the study. Suggest to delete.

Response: This has been revised done.

The rest of the paragraph shows that both providers and women had >90% satisfaction of care provided and is consistent. I am not sure how information provided on breastfeeding and family planning relates to RMC. This is more on comprehensiveness of care provided and not so much on RMC. Unless the women asked and were not provided with the information.

Response: This has been deleted as it does not add anything to the manuscript.

" return to the same facility". Would have been informative to see references or other studies that tested if there is association between RMC and future care preference or return to the same facility. Or not. I don't think this study was designed to test that association.

Response: We agree with this comment. We have revised the manuscript to reflect our inability to make this as a firm conclusion. We have indicated that the study was not moweded to measure psychometric properties on any indicator but we thought that, as the search for indicators to RMC is on-going, our study proposes that this could be one such indicators that will need testing. We have acknowledged the limitations and inherent confounding in the proposed indicator.

Discussion

Page 13. Line 49/50. "Many women were not allowed to express opinions in the care" were not allowed or were not comfortable. These are two different dimensions requiring different response.

Response: This whole paragraph has been revised to present the main findings of the study. The above statement has been removed completely.